# Online activities as risk factors for Problematic internet use among students in Bahir Dar University, North West Ethiopia: A hierarchical regression model

**Kerebih Asrese**⊙*, **Habtamu Muche**⊙

Social Work Department, Bahir Dar University, Bahir Dar, Ethiopia

* kerebih2000@yahoo.com

## Abstract

### Background

Problematic internet use (PIU) among youth has become a public health concern. Previous studies identified socio-demographic background risk factors for PIU. The effects of online activities on youth PIU behavior are not well investigated.

### Methods

This cross-sectional study assessed the roles of online activities for PIU behavior of undergraduate students in Bahir Dar University, North West Ethiopia. Data were collected from 812 randomly selected regular program students recruited from 10 departments. Respondents completed a pre-tested structured questionnaire. Hierarchical logistic regression models were used for analyses.

### Results

The results indicated that social networking (75.5%), entertainment (73.6%), academic works (70.9%), and online gaming (21.6%) are the important online activities students are engaging in the internet. About 33% and 1.8% of students showed symptoms of mild and severe PIU, respectively. Taking online activities into account improved the model explaining PIU behavior of students. Online activities explained 46% of the variance in PIU. Using the internet for social networking (AOR = 7.078; 95% CI: 3.913–12.804) and online gaming (AOR = 2.175; 95% CI: 1.419–3.335) were risk factors for PIU.

### Conclusions

The findings revealed that more than a third of the respondents showed symptoms of PIU. Online activities improved the model explaining PIU behavior of students. Thus, university authorities need to be aware of the prevalence of PIU and introduce regulatory mechanisms to limit the usage of potentially addictive online activities and promoting responsible use of the internet.

**Data Availability Statement:** The data underlying the results are attached as Supporting Information.

**Funding:** The authors received no specific funding for this work.

**Competing interests:** The authors have declared that no competing interests exist.

## Introduction

In the contemporary world, the internet is becoming an important part of peoples' daily life [1]. It is widely used in diverse areas of life such as education, academic activities and research, information exchange, interpersonal communication, commerce, science, and entertainment [2, 3]. Internet becomes available and affordable at homes, schools, colleges, libraries, and internet cafes [4]. Recently, the prevalence of internet users has increased rapidly, with the current estimated world's number of internet users in 2019 is 4.3 billion [5]. In Ethiopia, internet users increased from 360,000 in 2008 forming .43% of the population [5] to 16,437,811 internet users in 2019 forming 14.9% of the country's population [6].

While use of internet has enhanced the social and economic well-being of people [2], poor personal control over its use has become a concern [7] owing to increasing dependence on the internet for various aspects of lives [8]. Though debates on conceptualization and diagnosis are ongoing, the scientific community agreed that problematic use of digital technologies is associated with mental health problem [9].

Various terms and conceptualizations are used in the literature to label problematic use of digital technologies, such as "internet addiction" [10], "compulsive computer use" [11] or "problematic internet use" [12]. Though these competing definitions have various views and perspectives on the problem, the common concern for both is problematic use of digital technologies harms individuals [12]. For the sake of consistency, problematic internet use (PIU) defined as "excessive use of the internet that causes disturbances or harm to the individual" [12], is adopted in this study.

Problematic internet use is a compulsive behavior related to online activities in which individuals have inability to control internet usage despite its negative consequences [2]. It is reported that PIU leads to marked functional impairments in several aspects of life [13, 14], including social isolations [15, 16], unfriendly behavioral patterns [17, 18], attention deficit hyperactivity disorder [19], physical ill health [20], and performance and work difficulties [21].

PIU is a global phenomenon, especially among university students [22–24]. Possible reasons suggested for this are: universities provide free and unlimited access to the internet, students are away from parental control and without anyone monitoring what they do online, and students are encouraged by faculty members to use different internet applications [16]. In addition, the universities' settings foster a new student culture which necessitates the internet as a tool for communication, information sharing and community formation [25, 26].

With a broader education quality improvement initiative in higher institutions, the government of Ethiopia has deployed improved information communication technologies within the universities [27]. These institutions provide free and unlimited access to the internet; hence, students may use the internet more so than other population groups in the country [28]. Whether such context contributed for PIU behavior among university students in Ethiopia is not assessed.

To date, efforts to explain risk factors for PIU have identified a wide range of variables [29]. Being male [30, 31], adolescent ages [1], poor academic performance and family income [32–34], anxiety and depression [32], and low self-esteem [34–36], weak family support and low parental supervision [34, 37], peer pressure [38], and having free and easier internet access [33, 39] were noted contribute for developing PIU. Internet use variables as risk factors for PIU are also established. Age at first internet use [35] and online activities on the internet such as social networking [40–42] online gaming [31, 33, 42–44], frequency and length of internet use [45], online entertainments [46], and watching online pornography and online gambling [44] have been found more predictive of PIU.

The existing literature appeared that socio-demographic background factors and online activities people engaged in the internet are predictors of PIU. However, to the authors' knowledge, no research has been conducted taking into consideration all of them simultaneously. In addition, as most studies on risk factors have been conducted in overseas, our understanding of the risk factors for PIU in Ethiopian context is limited. Hence, in this study, we estimated the prevalence of PIU and examined socio-demographic background factors and online activity variables as potential risk factors for PIU. The findings may assist the prevention and management of PIU among students. Based on the existing literature, we hypothesized that the types of online activities in which students are engaged significantly predict PIU when socio-demographic background factors are controlled.

## Conceptual framework

We adopted Problem Behavior Theory [47] as a conceptual framework for the study. The theory is often used by researchers who investigated adolescents' misbehaviors [12]. Problem Behavior Theory specifies three fundamental systems: personality, perceived environment, and behavior. The background and socialization variables affect the personality and perceived environment systems and have a distal impact on behavior. The personality and perceived environment systems have proximal impact on the behavior [47].

In this study, similar variables were adopted. The students' socio-demographic background variables were grouped in one category as control variables and the online activity variables in other category as predictor variables and their hierarchical importance in predicting the PIU were assessed. Identifying the hierarchical importance of risk factors for PIU may help to design tailored intervention.

## Methods and materials

### Study setting, design and sample

An institution-based cross-sectional study was conducted in Bahir Dar University from February to March, 2018. The university has five campuses. Through education quality improvement initiative in higher institutions, free Wi-Fi is available in all campuses and private internet café services are also available around the campuses. At the time of the survey, the university enrolled more than 25,000 undergraduate students of which 14,884 are regular program students.

A single population proportion formula was used to estimate the sample size with 5% precision, 95% confidence, and a 10% non-response rate. There is no prior report about the estimated proportion of population with PIU in Ethiopian Universities. Therefore, we assumed this proportion to be 50% to increase the sample size of this study. The university is stratified by campuses and departments. And students in each department are stratified by years of study. We selected two departments randomly from each campus and students from year I to year IV were asked to participate in the study. To obtain representative sample, we used multistage stratified random sampling technique. Thus, to reduce the error that might incur during such multistage stratification, we multiplied the sample size by two (design effect). Therefore, the total sample size for the study was 844 students. Data obtained from 32 students were not complete, hence excluded.

The sample size was proportionally allocated to the selected departments, based on the number of regular students enrolled in the departments. Then systematic random sampling technique was used to select respondents from each selected department. Selected students were approached through their section mentors assigned by selected departments. The inclusion criteria included regular class undergraduate student, reside in the university dormitory,

used internet at least for six months, and not diagnosed with anxiety or depression disorders for a year and above.

## Study procedure

The study received ethical approval from Ethical Review Committee of Bahir Dar University. We initially developed the instrument in English and then translated into Amharic (national language) to ease understanding. Prior to the main study, a pretest was conducted on 25 regular undergraduate students (not included in the main survey) to check reliability and suitability of the instrument.

A week before data collection, we communicated selected departments with formal letter and we selected students who would participate in the study. Written consent of individual participants was obtained after being fully informed of the study purpose and procedures. It was also made clear to the respondents that participation was voluntary, and there would be no direct benefit or reward. We ensured confidentiality by removing all personal identities from the questionnaire. At each department, the questionnaires were self administered in a free lecture hall when students had free period. The investigators and two research assistants were available throughout the administration of the questionnaires to answer questions from individual students.

## Measurement

**Online activities.** In our exploratory qualitative study, we learned that students mainly use the internet for academic works, social networking, and entertainment. A few informants also shared that they exercise online gaming. Significant association between these online activities and PIU behavior was reported [48]. Thus, in this study, students were asked whether they have been engaging in the internet for these online activities, responded as yes/no.

**Socio-demography characteristics.** Socio-demographic background characteristics such as age, sex, year (time spent in the university), field of study, and academic performance were also collected. These attributes are reported have important role for PIU behavior [32, 49].

**Self-esteem.** The tool was developed to measure what respondents feel about themselves and what they think others think of them. It was adapted from Rosenberg [50] self-esteem test questions. The tool consisted of 10 items (sample items: On the whole, I am satisfied with myself; At times, I think I am not good at all) with four point responses ranging from 1 (strongly disagree) to 4 (strongly agree). The value ranges from 10 to 40, higher value indicating higher self-esteem. The internal consistency of the items in this study was good ($\alpha = 0.88$).

**Parental support.** The tool was adapted from Zimet and colleagues [51] developed to measure perceived social support. It was used to measure students' perception of their parents' support after they enrolled into the university. The instrument has seven items (sample items: I count on my parents' when things go wrong; I get the emotional help and support I need from my parents) with four point responses ranging from 1 (strongly disagree) to 4 (strongly agree). In this study, the items have good internal consistency ($\alpha = 0.93$). The score ranges from seven to 28, higher score indicating greater parental support.

**Peer pressure.** This tool was developed to assess the tendency of individuals to affiliate with like-minded friends. It was adapted from Steinberg and Monahan [52] developed to measure resistance to peer influence. The instrument has seven items (sample items: I often try what my friends do; I go along with my friends and do what they do to keep them happy of me) with four point responses ranging from 1 (strongly disagree) to 4 (strongly agree). The score ranges from seven to 28, with higher scores indicating greater peer pressure. The internal consistency of the items in this study was ($\alpha = 0.91$).

**Internet addiction test.** The 20-item Internet Addiction Test developed by Young [10] was adopted to measure the level of internet use in this study. The test has been widely adopted globally to measure the internet addiction levels of individuals [53]. In assessing the degree to which respondents' internet usage affected their daily routine, productivity, social life, psychological dependence, and time management [54], respondents were asked to rate items on a five-point Likert scale (1 = rarely and 5 = always). Item scores are added to create a final score between 20 and 100. Young suggested that a score of 20–49 points indicates an average online user who has complete control over their usage; a score of 50–79 reflects frequent problems due to Internet usage; and a score of 80–100 indicates that the internet is causing severe problems in the user's life [10]. The tool was pilot tested and the internal consistency of the items was very good ($\alpha = 0.93$).

**Data analysis.** All returned questionnaires were checked for completeness and consistency of responses manually. After cleaning, raw data were entered into SPSS for Windows versions 21 for analyses. Descriptive analysis was used to summarize the background characteristics of the respondents, their online activities, and to determine the prevalence of PIU. Between groups comparisons (non-PIU, mild PIU, and severe PIU) were performed using the Chi-square test of independence for categorical variables and one way ANOVA for continuous variables. Hierarchical multivariate logistic regression models were used to assess the relationship between independent variables and outcome variable. The socio-demographic background variables (controls) were entered into Model I. Model II added online activities variables (predictors) to Model I. Those variables with significant association to the outcome variable during bivariate analyses were entered during multivariate analyses.

Model I was nested into Model II. The overall fit of each logistic regression model was assessed by using its model Chi-square and goodness-of-fit indices (-2 log likelihood [-2LL]). We used model Chi-square and goodness-of-fit indices (-2 LL) to determine the improvement observed in Model II relative to Model I in explaining the dependent variable [18, 55, 56]. In addition to the indices of the overall model fit and model Chi-square, change in Nagelkerke's $R^2$ was evaluated as an approximate estimate of the amount of variance in the dependent variable accounted for by the models. Those variables with significant association to the dependent variable during bivariate analyses were entered into the multivariate logistic regression models. To test whether each individual factor had a significant relationship with PIU, Wald statistics were used. Multicollinearity among independent variables was checked using tolerance and variance inflation factor (VIF) values. The tolerance value ranged from 0.578 to 0.955 and the range of VIF was from 1.047 to 1.730.

## Ethical consideration

The study received ethical approval from Ethical Review Committee of Bahir Dar University. A week before data collection, we communicated selected departments with formal letter and we selected students who would participate in the study. Written consent of individual participants was obtained after being fully informed of the study purpose and procedures. It was also made clear to the participants that participation was voluntary, and there would be no direct benefit or reward. We ensured confidentiality by removing all personal identities from the questionnaire.

## Results

### Background characteristics of respondents

Eight hundred twelve students (66% males and 34% females) participated in the study. Respondents' age ranged from 18 to 27 years old with mean age of 21.38 years. Students' fields

of studies were social sciences (36%), law and Land Administration (21.6%), engineering (19%), textile (16%), and agriculture (7%). Majority of the respondents (30%) were 2<sup>nd</sup> year students followed by 3<sup>rd</sup> year (25%) and fourth year and freshman (22%). The four most important online activities of students were social networking (75.5%), entertainment (watching videos, sports, music, and news) (73.6%), academic works (72.5%), and online gaming (61%). Respondents' cumulative grade point average ranged from 1.84 to 3.92 with mean grade point average 3.05. Respondents' means scores on parental supervision, self-esteem, and peer pressure were 20.07, 30.84, and 12.95, respectively (Table 1).

## Prevalence of internet addiction

In this study, the prevalence of sever PIU was 1.8% (95% CI: 1% - 2.8%) and mild PIU was 33.4% (95% CI: 29.9% - 36.5%) (Table 1). The prevalence rate including mild and severe PIU

**Table 1.** Distribution of respondents by background characteristics, online activities, and internet use status (n = 812).

| Background characteristics | n (%) |
|---|---|
| **Sex** | |
| Male | 535 (65.9) |
| Female | 277 (34.1) |
| **Use the internet first** | |
| Secondary school | 567 (69.8) |
| At university | 135 (16.8) |
| At elementary school | 110 (13.5) |
| **Field of study** | |
| Social sciences | 293 (36.1) |
| Engineering | 154 (19) |
| Law and land administration | 175 (21.6) |
| Agriculture | 59 (7.3) |
| Textile | 131 (16.1) |
| **Years spent at the university** | |
| 1<sup>st</sup> year | 177 (21.8) |
| 2<sup>nd</sup> year | 247 (30.4) |
| 3<sup>rd</sup> year | 207 (25.5) |
| 4<sup>th</sup> year and above | 181 (22.3) |
| Social networking use | 613 (75.5) |
| Entertainment use | 598 (73.6) |
| Academic works use | 589 (70.9) |
| Online game use | 175 (21.6) |
| **Age** | 21.38 (1.70)* |
| **Cumulative grade point average** | 3.05 (.45)* |
| **Parental supervision** | 20.07 (5.15)* |
| **Self-esteem** | 30.84 (5.52)* |
| **Peer pressure** | 12.95 (4.13)* |
| **Internet use status** | |
| Non- PIU | 526 (64.8) |
| Mild PIU | 271 (33.4) |
| Severe PIU | 15 (1.8) |

* = mean (standard deviation).

was significantly higher for females than males (p < .05). Statistically significant relationship was found between years spent at the university and PIU indicating that senior students were more likely to have symptoms of PIU than juniors (p < .001). Greater proportions of students using the internet for social networking and for online gaming were more likely to have symptoms of PIU than their counterparts (p < .001). Lower proportion of students using the internet for academic works than those who did not use the medium for same purpose had PIU behaviors (p < .001). The one-way ANOVA results also demonstrated that the groups significantly differ in age (F (2,809) = 5.53, p < 0.05), grade point averages (F (2,809) = 65.67, p < 0.001), parental supervision (F (2,809) = 12.38, p < .001), self-esteem (F (2,809) = 38.37, p < .001), peer pressure (F (2,809) = 67.06, p < 0.001), and internet use score (F (2,809) = 145, P < .001) (Table 2).

## Risk factors for internet addiction

Table 3 presents the results of multivariate logistic regression analyses of the association between internet use status and various socio-demographic background and online activities of respondents. Because severe PIU rate was very low, we combined severe PIU and mild PIU into one group and computed the contributors of PIU based on two groups of respondents which were non-PIU and PIU users. As illustrated in the Table, the two models were significantly associated with PIU. Model II had smaller -2LL value than Model I (724.059 vs. 802.109) indicating that the inclusion of online activity variables significantly improved the goodness-of-fit of Model II as compared to Model I ($\chi^2$ (11) = 329.605, P <0.001). While Model I accounted for 36.70% of the variance in PIU (Nagelkerke $R^2$ = .367), Model II accounted for 45.90% of the variance in PIU (Nagelkerke $R^2$ = .459). The results indicated that there is statistically significant improvement in predicting PIU of students with the online activity variables after controlling the socio-demographic background variables.

Regarding online activity variables, use of internet for social networking was the strongest predictor of PIU. Students using the internet for social networking were more likely to have PIU than those who did not use the medium for such purpose (AOR = 7.078, 95% CI: 3.913–12.804). Compared to students who were not using the internet for online gaming, students using the medium for online gaming were more likely to be internet addicted (AOR = 2.175, 95% CI: 1.419–3.335) (Table 3).

The findings also indicated that socio-demographic background variables- seniority in the university, self-esteem, academic performance, parental support, and peer pressure were significant risk factors for PIU. Junior students were less likely to have symptoms of PIU behavior than senior students. With one point increase in grade point average students were .193 (95% CI: .125–.298, p < 0.001) times less likely to have symptoms of PIU. Increasing self-esteem (AOR = .948, 95% CI: .912-.985) and parental support (AOR = .954, 95% CI: .919-.990) were protective of PIU. On the other hand, students with increasing peer pressure were more likely to have symptoms of PIU (AOR = 1.133, 95% CI: 1.075–1.174) (Table 3).

## Discussion

This study investigated the roles of online activities on PIU behaviors in a sample of undergraduate regular program students in Bahir Dar University. In this sample of students, 33.4% and 1.8% were classified as having mild PIU and severe PIU, respectively. Considering the similarity of measurement tools used, the results are comparable to the rates of 37.1% and 2.9% reported among university students in Jordan [57]. The results in the current study are higher than the rates of 10.4% and 0.8% reported among medical college students in India [58]. On the other hand, the results in this study are lower than the rates of 59.2% and 6.5% reported in

**Table 2. Respondents' socio-economic background characteristics and online activities by internet use status (n = 812).**

| Background characteristics | Internet use status | | | $\chi^2$ | Multiple comparisons |
|---|---|---|---|---|---|
| | Non-PIU n (%) | Mild PIU n (%) | Severe PIU n (%) | | |
| Sex | | | | 7.88* | |
| Male | 335 (66.4) | 175 (32.7) | 5 (.9) | | |
| Female | 171 (61.7) | 96 (34.7) | 10 (3.6) | | |
| Use the internet first | | | | .79 | |
| Before university | 436 (64.4) | 229 (33.8) | 12 (1.8) | | |
| After university | 90 (66.7) | 42 (31.1) | 3 (2.2) | | |
| Field of study | | | | 15.12 | |
| Social sciences | 192 (65.5) | 97 (33.1) | 4 (1.4) | | |
| Engineering | 92 (59.7) | 61 (39.6) | 1 (.6) | | |
| Law | 123 (70.3) | 48 (27.4) | 4 (2.3) | | |
| Agriculture | 42 (71.7) | 17 (28.8) | - | | |
| Textile | 77 (58.8) | 48 (36.9) | 6 (4.6) | | |
| Years spent at the university | | | | 42.72*** | |
| 1st year | 147 (83.1) | 28 (15.8) | 2 (1.1) | | |
| 2nd year | 161 (65.2) | 84 (34) | 2 (.8) | | |
| 3rd year | 119 (57.5) | 84 (40.6) | 4 (1.9) | | |
| 4th year and above | 99 (54.7) | 75 (41.4) | 7 (3.9) | | |
| Social networking use | 343 (56) | 255 (41.6) | 15 (2.4) | 85.63*** | |
| No use | 183 (92) | 16 (8) | - | | |
| Entertainment use | 390 (65.2) | 200 (33.4) | 8 (1.3) | 3.26 | |
| No use | 136 (63.6) | 71 (33.2) | 7 (3.3) | | |
| Academic works use | 404 (70.1) | 163 (28.3) | 9 (1.6) | 24.96*** | |
| No use | 122 (51.7) | 108 (45.8) | 6 (2.5) | | |
| Online game use | 82 (46.9) | 81 (46.3) | 12 (6.9) | 52.51*** | |
| No use | 444 (69.7) | 190 (29.8) | 3 (.5) | | |
| | | | | F-test | |
| Age[a] | 21.23 (1.76) | 21.64 (1.6) | 21.80 (1.14) | 5.53* | NPIU vs. MPIU* |
| Grade point average[a] | 3.17 (.42) | 2.86 (.42) | 2.42 (.25) | 65.67*** | NPIU vs. MPIU; vs. SPIU*** MPIU vs. SPIU*** |
| Parental supervision[a] | 20.67 (4.95) | 19.10 (5.8) | 16.46 (5.86) | 12.38*** | NPIU vs. MPIU; vs. SPIU** |
| Self-esteem[a] | 31.96 (4.96) | 29.01 (5.6) | 24.66 (7.06) | 38.37*** | NPIU vs. MPIU; vs. SPIU*** MPIU vs. SPIU** |
| Peer pressure[a] | 11.88 (3.44) | 14.70 (4.1) | 18.93 (5.35) | 67.05*** | NPIU vs. MPIU; vs. SPIU*** MPIU vs. SPIU*** |
| Sum of internet use | 36.66 (7.80) | 61.06 (8.2) | 84.06 (3.06) | 145.70*** | NPIU vs. MPIU; vs. SPIU*** MPIU vs. SPIU*** |
| **Total** | **526 (64.8)** | **271 (33.4)** | **15 (1.8)** | | |

NPIU = non-PIU, MPIU = mild PIU, SPIU = severe PIU

[a] = mean (standard deviation)

*p < .05

**p < .01

***p< .001.

Namibia and the rates of 70.3% and 4.7% reported in Uganda university students [59] and the rates of 56.5% and 7.8% reported for Malaysian undergraduate students [60]. The rate of severe PIU in the current study is also lower than the rate of 10.2% among Nigerian [61] university

**Table 3. Hierarchical Logistic Regression analysis of students' socio-demographic background characteristics and their online activities predicting PIU (N = 812).**

| Variables | Model I | | | | Model II | | | |
|---|---|---|---|---|---|---|---|---|
| | B | S.E. | Wald | AOR (95% CI) | B | S.E. | Wald | AOR (95% CI) |
| Sex | | | | | | | | |
| Male | .092 | .202 | .209 | 1.097 (.738–1.631) | -.005 | .218 | .001 | .995 (.649–1.524) |
| Female[R] | | | | 1 | | | | 1 |
| Year of study | | | | | | | | |
| 1st year | -2.253 | .375 | 36.184*** | .105 (.050-.219) | -2.060 | .399 | 26.699*** | .127 (.058-.278) |
| 2nd year | -.766 | .273 | 10.263** | .417 (.244-.712) | -.699 | .292 | 5.719* | .497 (.280-.882) |
| 3rd year | -.623 | .257 | 5.875* | .536 (.324-.888) | -.597 | .272 | 4.828* | .550 (.323-.938) |
| 4th year and above[R] | | | | 1 | | | | 1 |
| Age | -.017 | .070 | .056 | .984 (.858-1.28) | .037 | .076 | .241 | 1.038 (.895-1.2.44) |
| Grade point average | -1.732 | .215 | 65.037*** | .177 (.116-.270) | -.1.645 | .226 | 53.234*** | .193 (.124-.301) |
| Self-esteem | -.051 | .019 | 7.436** | .951 (.917-.986) | -.053 | .020 | 7.209** | .948 (.912-.986) |
| Parental support | -.043 | .017 | 6.239* | .957 (.925-.991) | -.047 | .019 | 6.202* | .954 (.919-.990) |
| Peer pressure | .152 | .025 | 36.677*** | 1.167 (1.108–1.223) | .125 | .027 | 21.632*** | 1.133 (1.075–1.194) |
| Social networking use | | | | | 1.957 | .303 | 41.825*** | 7.080 (3.912–12.813) |
| Academic works use | | | | | -.290 | .206 | 1.975 | .748 (.500–1.121) |
| Online gamind | | | | | .777 | .218 | 12.692*** | 2.176 (1.419–3.337) |
| -2LL | 801.900 | | | | 724.059 | | | |
| Model $\chi^2$ | 251.764*** | | | | 329.606*** | | | |
| Degree of freedom | 9 | | | | 12 | | | |
| Nagelkerke $R^2$ | .367 | | | | .459 | | | |
| $\Delta R^2$ | - | | | | .092 | | | |

[R] = reference category

*p < .05

**p < .01

***p < .001, AOR (95% CI) = Adjusted odd ratio (95% confidence interval)- net effect of each independent variable on the dependent variables, $\Delta R^2$ = change in Nagelkerke $R^2$.

students. Variations in the prevalence of PIU could be due to cultural diversity among communities and the time frame when the research was conducted [30, 32]. Even so, the findings in our study has demonstrated that PIU is relatively higher in Ethiopia despite internet penetration is much more limited than other countries [27].

As we hypothesized, the analyses of factors for PIU revealed that students' online activities significantly predicted students' PIU behaviors when socio-demographic background variables are controlled. Compared to the Model that included socio-demographic background variables, the inclusion of online activities improved the fit of the Model predicting PIU behavior. The results corroborate previous studies that online activities people engaged in the internet are risk factors for internet addiction [33, 42, 62, 63].

Of all the online applications examined in this study, social networking was the strongest predictor of PIU. The finding corroborates previous research [40–42] reporting that engagement in online social networking is most important risk factor for PIU. Social networking applications are used mainly to maintain and establish offline and online networks [40] that their excessive uses has a variety of negative consequences for the individual [63].

The other Internet online application that significantly increased the risks of PIU in this study was online gaming. Playing online games increased the risk of PIU on the internet by 117.5%. This finding is consistent to the existing literature [33, 42–44, 63] documenting

positive association between online gaming and PIU. Online gaming has been identified as potentially addictive as it requires large amount of commitment and time investment from the player which may in turn contribute to the development of maladaptive behaviors that reinforce gaming [64]. Our findings that engagements in social networking and online gaming have been increased the risk of PIU on the internet highlighted that the usage of some online activities are potentially problematic as overuse can result in a variety of negative consequences [40]. Thus, university officials and other concerned bodies may put regulatory mechanisms in place to limit the usage of potentially problematic internet applications such as social networking and online gaming.

This study also demonstrated that socio-demographic background variables such as self-esteem, academic performance, and number of years stayed in the university (seniority), parental support, and peer pressure significantly associated with PIU. The current finding revealed that higher level of self-esteem was protective of PIU. The result supports previous studies [34, 36, 62, 65] reporting that lower self-esteem was related to PIU. Consistent with previous studies [23, 31, 32], the finding in this study revealed a statistically significant negative relationship between academic grade point average and PIU. Gencer and Koc [66] noted that students with poor academic performance may experience stress and may develop low self-esteem; therefore, they use the internet as a way to cope these stressors. Others [16] argued that individuals with PIU often experienced lack of sleep since they stay awake during late-night hours in order to surf through various web pages. The lack of sleep causes lack of concentration and loss of interest in everyday lectures, resulting in reduced reading of the course material and eventually lower marks at exams. Our findings did not indicate whether poor academic performance or PIU is precursor for the observed relationship. Therefore, future longitudinal study is called for uncovering such causal relationships.

In this study, the proportion of students with PIU sequentially increased with increasing years spent in the university (freshman to fourth year). This finding is consistent with previous studies in Nigeria [61] reporting the likelihood of internet addiction among university students vary with respect to year of study in that senior students were more likely to show PIU symptoms than their juniors. This could be attributed to the fact that parents might have limited support for senior students. Students who successfully accomplished university's expectations in each year will be seniors in turns. These students may have sufficed time to use internet excessively for different purposes. On the other hand, as seniority increases (second year and above), students may be expected to accomplish difficult tasks accordingly (e.g., assignments and research projects) that might cause stress, and students might use the internet as a method to cope the context [16]. Though these explanations are intuitive, the current result that senior students are significantly more likely to PIU behavior than their junior counterparts is informative for the university administrators to design interventions to inculcate responsive internet use behavior at the beginning of students' university life.

This study identified inverse relationship between parental support and students' PIU. Such negative association between parental support and adolescents' PIU was repeatedly reported in studies conducted in different communities [33, 67, 68]. Good social support of families and friends buffers from psychological stress in individuals [69] and stimulates an individual to improve one's perceived efficacy to lessen the negative consequences of the stressful experience [70]. Wu et al. also suggested that lack of social support is both the cause and consequence of PIU [33]. Hence, consolidating familial ties and promoting helpful relationships might prevent PIU in adolescents [71].

On the other hand, positive significant relationship was found between peer pressure and the risk of PIU. Every unit increase in peer pressure score increased the odds of PIU risk by16%. Social norm theory assumes that adolescents' beliefs about the norms that are prevalent

among their peers influence their behavior [72] through descriptive and injunctive peer norms [73]. Descriptive peer norms are adolescents' perceptions about the quantity and frequency of a certain behavior among peers. Injunctive peer norms are beliefs about the approval of a behavior among peers [73]. In this study, it is not clear whether the students' perceptions of their peers' internet use behavior or their approval of internet use that serve as factors for PIU. Regardless of the direction, the current findings highlighted the importance of peer pressure in students' PIU behavior. Thus, programs designed to reduce PIU behavior may focus on students' perception of normative behavior.

## Limitations

This study has limitations that should be considered. We assessed internet use status of students using IAT, the most commonly used measure of Internet addiction [74]. Though strong internal reliability estimates of IAT have been established [75], researchers have reported different factor structures [76] which suggests a potential lack of construct validity of the instrument [74]. The absence of evaluation of the validity of the cut-off scores also limits its usefulness as a potential screening tool [77]. Some items also have been considered outdated or vague as that need to be removed or improved [78]. Future research may assess the concurrent validity of IAT with recent tools.

Of the 844 students, 812 students (96.21% response rate) fully completed the questionnaire. The participants were obtained from a single government university which might not be representative of the entire university students in Ethiopia. Data were collected using self administered structured questionnaire. Students may give responses which they believed to be expected or acceptable, thus there might be measurement bias. We also collected data only from students who participated in the study. The lack of data from significant others (e.g., parents and network members) is the limitation of the study. In this study, the purposes of internet use (students' engagement in online activities) were measured categorically as yes/no. Such binary scoring did not indicate the amount of internet use (temporal dimensions) hence likely to lead to inflated prevalence of each activity. Future research that will consider the time spent online of each activity within a day/ a week will yield better evidence. Also, students may engage in the internet for more than one purpose or some online activities may have combined purposes. Such combined uses may be the greatest risk for PIU. Future research may discover such mingled proofs. Being informed by the exploratory qualitative results, we did not assess watching online pornography and online gambling activities as potential risks for PIU. Previous study [31, 46] reported that these online activities are risks for PIU. Hence, future research may incorporate these variables to understand what kinds of internet use cause the greatest risk.

In addition, the cross-sectional nature of the study limited the interpretation of the findings in terms of cause-effect relationships. There are also many factors this study did not assess, including environmental influences (university settings) and intrapersonal influences (e.g., anxiety, depression, and substance abuse). Future research may attempt to address these factors into consideration to predict risk behaviors for students' PIU. Studies on factors contributing to responsible internet use behavior are also needed to assess the assets in and outside students so that interventions and programs which can foster such behavior can be developed and implemented.

## Conclusion

This study assessed the influence of online activities as risk factors for PIU among undergraduate regular program students in Bahir Dar University. The findings indicated that social

networking, entertainment, academic works, and online gaming are important activities students are doing online. More than a third of the students (35.2%) showed symptoms of PIU. The hierarchical logistic regression results revealed that students' engagement into online activities improved the model explaining the PIU when socio-demographic background variables are controlled. Students who are engaging in the internet for social applications such as social networking and online gaming were more likely to show PIU. The findings in this study are important and timely as the internet has become the primary medium for information access in our universities. Thus, university authorities need to be aware of the prevalence of PIU and its antecedents so that interventions can be designed to prevent adverse outcomes. Interventions should focus on identifying students with PIU, creating awareness on the its negative effects, counseling services to develop students' self image, introduce a regulatory mechanisms to limit the usage of potentially problematic internet applications, and promoting responsible use of the internet at the beginning of students' university life.

## Supporting information

**S1 File. Survey questionnaire.**
(DOCX)

**S2 File. Survey data.**
(SAV)

## Acknowledgments

The authors would like to thank students who generously gave their time to complete the questionnaire. We would like also acknowledge the deans of the faculties who facilitated our data collection.

## Author Contributions

**Conceptualization:** Kerebih Asrese, Habtamu Muche.

**Data curation:** Habtamu Muche.

**Formal analysis:** Kerebih Asrese.

**Investigation:** Kerebih Asrese.

**Methodology:** Kerebih Asrese, Habtamu Muche.

**Supervision:** Habtamu Muche.

**Writing – original draft:** Kerebih Asrese, Habtamu Muche.

**Writing – review & editing:** Kerebih Asrese.

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
