## [Decision Letter · Decision Letter 0]

17 Mar 2020

PONE-D-20-00961

Online activities as risk factors for internet addiction among students in Bahir Dar University, North West Ethiopia: Hierarchical regression model

PLOS ONE

Dear Dr. Asrese,

Thank you for submitting your manuscript to PLOS ONE. After careful consideration, we feel that it has merit but does not fully meet PLOS ONE’s publication criteria as it currently stands. Therefore, we invite you to submit a revised version of the manuscript that addresses the points raised during the review process.

We have received three excellent reviews that you should follow carefully. In general, the manuscript requires to update the literature and a much more careful statistical analysis. Moreover, some definitions are not clear enough and the language must be revised in detail.Please follow the comments raised by the reviewers. 

We would appreciate receiving your revised manuscript by May 01 2020 11:59PM. To enhance the reproducibility of your results, we recommend that if applicable you deposit your laboratory protocols in protocols.io, where a protocol can be assigned its own identifier (DOI) such that it can be cited independently in the future. For instructions see: http://journals.plos.org/plosone/s/submission-guidelines#loc-laboratory-protocols

We look forward to receiving your revised manuscript.

Kind regards,

Alfonso Rosa Garcia

Academic Editor

PLOS ONE

Journal Requirements:

2. We noticed you have some minor occurrence of overlapping text with the following previous publications, which needs to be addressed:

* Malak, Malakeh Z., Anas H. Khalifeh, and Ahmed H. Shuhaiber. "Prevalence of Internet Addiction and associated risk factors in Jordanian school students." Computers in Human Behavior 70 (2017): 556-563.

* Frangos, Christos C., Constantinos C. Frangos, and Ioannis Sotiropoulos. "Problematic internet use among Greek university students: an ordinal logistic regression with risk factors of negative psychological beliefs, pornographic sites, and online games." Cyberpsychology, Behavior, and Social Networking 14.1-2 (2011): 51-58.

In your revision ensure you cite all your sources (including your own works), and quote or rephrase any duplicated text outside the methods section. Further consideration is dependent on these concerns being addressed.

4. Please provide your institutional email address.

Reviewers' comments:

Reviewer's Responses to Questions

**Comments to the Author**

1. Is the manuscript technically sound, and do the data support the conclusions?

Reviewer #1: Partly

Reviewer #2: Yes

Reviewer #3: Partly

2. Has the statistical analysis been performed appropriately and rigorously? 

Reviewer #1: Yes

Reviewer #2: No

Reviewer #3: Yes

3. Have the authors made all data underlying the findings in their manuscript fully available?

Reviewer #1: Yes

Reviewer #2: No

Reviewer #3: No

4. Is the manuscript presented in an intelligible fashion and written in standard English?

Reviewer #1: Yes

Reviewer #2: Yes

Reviewer #3: Yes

5. Review Comments to the Author

Reviewer #1: Thank you for the opportunity to review this manuscript on Internet addiction in Ethiopian university students. I have the following specific comments:

1. The introduction does not provide strong enough a reason to put individual-level, environmental, and online activities all into a single model for the prediction of Internet addiction. In theory, if each group of these variables could explain part of the variance in Internet addiction risks, it is acceptable to estimate them in separate models unless there are other causal relationships between them. The authors should explain why they think results may be different if they are put into the same model compared with separately estimated. Otherwise, there does not appear to be any new knowledge to anticipate judging from the introduction.

2. Also, although not all variables were considered, this study has tried to estimate a comprehensive explanatory model containing most of the variables you have mentioned for predicting Internet addiction: https://doi.org/10.1016/j.chb.2016.11.021

3. The method of calculation for the required sample size seems to be one used for prevalence studies (assuming 50%). I do not think this study is one of those. Authors may consider justifying their sample size using another method.

4. Was the Ethiopian IAT validated? If not, that is a limitation and needs to be mentioned.

5. How were the online activities reported? Frequency? Perceived amount of use? Need to state these more clearly. And all levels of all categorical variables being analyzed need to be explicitly stated.

6. In Table 1 it seems online activities are simply dichotomized as yes and no, and that is quite simplistic. I think a more specific frequency of use, e.g. once a week, twice a week,… for the particular online activities would capture more information than this.

7. The following statement is an incorrect interpretation of the result. You cannot say something is more important than the existing variables because it improves the fit of the model. I am sure if you put online activities into the model first, then add individual attributes and environmental factors the model fit would improve as well.

“Thus, the results revealed that students’ online activity variables are more risk factors for IA than their individual attributes and environmental factors.”

8. The authors need to understand that they didn’t ‘examined the relative importance’ of risk factors of IA. Interpretation needs to be revised thoroughly.

Reviewer #2: This investigated the relative contribution of the individual, environmental and content to problematic internet use.

Although it is an interesting study but a major issue is found for the statistical analysis. The authors must provide more details whether the required conditions were met for multiple regression.

Table 1

I strongly recommend the authors to reorganize the left column of the table to a single row to improve the readability.

e.g.

Online game -> Online game use

Use

Page 13

Better to present the significant digit in a unified way

32 – 38.7 -> 32.0-38.7

I think providing the t-score or chi-square number is too much of unneccessay information and suggest their omission from the text.

e internet addicted; what do you mean by e internet ?

Moreover, labeling addiction just by high scores on self-test score could be regarded as overstatement. I suggest changing the term to problematic internet use (PIU) instead of IA.

Compared to non-addicted, addicted students were significantly older (t (810) = -3.31, p < .05), had lower grade point averages (t (810) = 10.65, p < .001), had lower parental supervision (t (810) = 4.57, p < .001), had lower self-esteem (t (810) = 8.15, p < .001), and experience higher peer pressure

I think these are the main findings and should be highlighted by being included in the abstract!

When examing the risk factors by logistic regression, please add further explanation.

parental supervision or self-esteem

R2 = .459 -> 0.46 significant digit in a unified way; i.e. either three or two digits rather than mixture of both with exemption to p-value)

(AOR = 7.01, 95% CI: 290 3.91 – 12.083.91 – 12.08) ; seems like error. Please check!

p. 16

students were .19 (95% CI: .12 –.29, p < 0.001) times less likely to be internet addicted and with one point increase in self-esteem score students were .95 times (95% CI: .92 - .98, p < .01)

Please be consistent with the display of digits and I recommend against omitting zero. This applies to others for number presenations like Table 3.

one point increase in self-esteem

one point increase in parental supervision

In reality, self-esteem or parental supervision is not a subject to be actually manipulated by ‘one’ point. Please consider using plain words such as higher self-esteem or greater parental supervision.

Table 3

I again recommend the authors to reorganize the left column of the table to a single row to improve the readability.

Is sex included for the model I analysis?

University year and age may overlap. Please provide more data for collinearity check.

The Nagelkerke R2 for model I was 0.306, 0.367 for the model II and 0.459 for the model III.

I can see the difference between II and III (0.092) is larger than the difference between I and II (0.061). However, the R2 for model I was 0.306 in the first place, which is way more bigger than 0.092. Therefore, ‘Online activities better predict students’ addicted internet use behavior than individual and environmental attributes’ is a misleading statement. The data presented by the authors suggests contribution by the following orders; individual > online activity types > environment.

Reviewer #3: The manuscript presents an original study focused on (adolescent) Internet addiction (IA), which is the important and increasingly prevalent topic. The main aim was to analyze the effect of possible predictors (engagement in specific online activities, the amount of years spent at university, self-esteem and others) on IA in Ethiopian university students. The aim and study design are sound, but there are some issues that should be addressed before publishing the study. After addressing these issues, the study could be a valuable contribution to current knowledge, especially because it presents data on Internet addiction from Africa, which are currently scarce (the most studies within the area came from Asia, then Southern and Western Europe and Northern America).

Major issues

Introduction:

Introduction should be more focused on the main hypothesis, i.e., the effect of online activities on IA. Some sources dealing with this issue are missing. You might want to check at least these two:

Siomos, K., Floros, G., Fisoun, V., Evaggelia, D., Farkonas, N., Sergentani, E., Lamprou, M., & Geroukalis, D. (2012). Evolution of Internet addiction in Greek adolescent students over a two-year period: The impact of parental bonding. European Child & Adolescent Psychiatry, 21(4), 211–219. https://doi.org/10.1007/s00787-012-0254-0

Wu, X.-S., Zhang, Z.-H., Zhao, F., Wang, W.-J., Li, Y.-F., Bi, L., Qian, Z.-Z., Lu, S.-S., Feng, F., Hu, C.-Y., Gong, F.-F., & Sun, Y.-H. (2016). Prevalence of Internet addiction and its association with social support and other related factors among adolescents in China. Journal of Adolescence, 52, 103–111. https://doi.org/10.1016/j.adolescence.2016.07.012

Generally, the introduction is not very up to date with most sources published before 2015 (or even 2012). Given to the quick pace of scientific development within the area, this is a significant drawback. So please be sure to update your introduction thoroughly.

Few examples:

- Row 81: While defining IA, maybe you could use more relevant source or source with greater impact (from leading experts in the field) – see e.g. Fineberg, N., Demetrovics, Z., Stein, D., Ioannidis, K., Potenza, M., Grünblatt, E., Brand, M., Billieux, J., Carmi, L., King, D., Grant, J., Yücel, M., Dell’Osso, B., Rumpf, H., Hall, N., Hollander, E., Goudriaan, A., Menchon, J., Zohar, J., … Chamberlain, S. (2018). Manifesto for a European research network into Problematic Usage of the Internet. European Neuropsychopharmacology, 28(11), 1232–1246. https://doi.org/10.1016/j.euroneuro.2018.08.004 and Kuss, D. J., & Billieux, J. (2017). Technological addictions: Conceptualisation, measurement, etiology and treatment. Addictive Behaviors, 64, 231–233. https://doi.org/10.1016/j.addbeh.2016.04.005

- Row 81-84: While listing the negative outcomes of IA (which concerns your own study only marginally), it would be better to rely on available literature review or meta-analytic studies on the negative effects of IA (or similar concepts – e.g., Internet Gaming Disorder).

- Row 99-100: There are some recent and relevant sources concerning the effect of parental support and parental control on Internet addiction, that were not mentioned.

Aims:

The authors claimed they want to analyze whether the type of online activity (resp. the purpose of using internet) is better (stronger) predictor of IA than individual and environmental attributes (row 111-112). I find this to be problematic for several reasons.

- First, it is not possible to directly compare strength of predictors by hierarchical regression – you can only observe, whether predictors included in the initial model remain significant after the inclusion of other predictors; and whether new predictors enhance the overall explanatory power of the model. I would recommend reformulating the hypothesis in a manner similar to this: “The type of online activity is significant predictor of IA, even when other background variables (namely ….) are included.” If you really want to compare the strength of predictors, you should test two models (one with the main predicting variable – the online activities) and second with all other predicting variables and then compared their fit and explanatory power.

- Second, you claim to assess students’ individual and environmental attributes. This sounds very complex, but in fact you assessed some common sociodemographic variables (sex, age, field of study, year of study) and few other variables (grades, self-esteem, perceived parental support and peer pressure resistance). I do not claim that It is bad, it is similar to what we can find in other studies, but it is not so complex and comprehensive how implied by very general term “individual and environmental attributes”. Moreover, I am not sure whether it is possible to categorize your variables as being strictly “individual” or “environmental” (e.g., is parental support as perceived by an adolescent more environmental or individual?). I would recommend omitting the individual/environmental categories completely (from the whole manuscript, including results, table captions etc.). Instead, please clearly state what are your main predicting variables (online activities?) and what are the controlling (background) variables.

- Third, the term “internet addicted behavior” is not commonly used, I would recommend using “Internet addiction” or “addictive use” or “problematic internet use” throughout the whole manuscript.

Methods:

The design is described relatively clearly and in detail, but there are few points that should be addressed.

- First, when planning the sample size, how come you had expected IA to be prevalent in 50% of university students, when the worldwide prevalence is much lower? Or did I misunderstand? (row 124-125)

- Second, in two of your measures I see the obvious discrepancy between conceptualization (what you claimed to assess) and what you really assessed: (1) “parental supervision” is completely different phenomenon than “parental support”. They also have different associations with IA. Please consult the literature on parenting, parental control (demandingness, strictness) and parental warmth (responsiveness, support) and make sure that you use terms well. (2) You were using the term “peer pressure,” but measured “resistance to peer pressure” – this may be quite confusing, while these are basically “reversed” concepts, if I am reading it right.

- Third, could you specify whether you used the common cut-offs for Young’s Internet addiction test scores? You should be extra careful in this respect, while you are talking about IA prevalence (more about it also further).

- Fourth, you did not present the response rate and the amount of missing values and how they were addressed. Could you add this important information to the manuscript, please?

Results:

Results are quite clear and well-organized. Tables are standard. There are few points that should be addressed:

- First, there are significant associations between IA and both the age and years spent at university. While these two variables are most probably also associated, one of the two associations with IA could be artificial. Could you add another analysis that would shed light upon this? Or at least you should elaborate this while discussing these results.

- Second, when comparing predictive models (page 15), you should report not only R2 and its changes, but also F-value (with significance), that shows how much (and if significantly) models differed, especially when you stated that “there is statistically significant improvement in predicting IA”(row 283). The analysis does not imply the conclusion you made (row 285-286), it only showed that including the type of online activity significantly improved the prediction of IA. So please correct this (it relates to the point about hypothesis that I have made earlier).

- Third, the association between grades and IA (row 295-298). The impaired academic performance is a recognized negative outcome of IA. Therefore, it does not make much sense to test the reverse causality hypothesis without the supporting theory. However, you mentioned the theory in discussion – maybe you could elaborate it in introduction and perform the analysis with self-esteem as a mediating variable between grades and IA.

Discussion:

The discussion is standardly structured (the summary of results, discussion of results), but the content has several limitations.

- (1) The sources are not very recent and sometimes I missed the logic behind their inclusion – e.g., when discussing prevalence of IA, you reported prevalence in Hong Kong and Greece, but there are relatively recent paper with worldwide prevalence - e.g. Cheng, C., & Li, A. Y. (2014). Internet Addiction Prevalence and Quality of (Real) Life: A Meta-Analysis of 31 Nations Across Seven World Regions. Cyberpsychology, Behavior, and Social Networking, 17(12), 755–760. https://doi.org/10.1089/cyber.2014.0317. Moreover, the “Hong Kong” paper actually estimated the prevalence in six Asian countries… could you please clarify whether it is really prevalence for Hong Kong? And if so, then why you picked this one country?

- (2) Moreover, when discussing the prevalence, you should take into account the measurement of IA. You used screening test and based the prevalence on the presence of even “mild symptoms” of IA, which may be different approach then used in other studies. Please make the thorough comparison of your approach with your key source [46].

- (3) Discuss your results on the type of online activity with relevant studies (at least those two mentioned earlier in this review: Siomos et al., 2012; Wu et al., 2016).

- (4) Please be careful about the causality statements, especially in case of grades (as pointed out earlier) (row 343-345).

- (5) There are more recent sources dealing with IA and self-esteem. Please check at the very least this one: Dong, B., Zhao, F., Wu, X.-S., Wang, W.-J., Li, Y.-F., Zhang, Z.-H., & Sun, Y.-H. (2019). Social Anxiety May Modify the Relationship Between Internet Addiction and Its Determining Factors in Chinese Adolescents. International Journal of Mental Health and Addiction, 17(6), 1508–1520. https://doi.org/10.1007/s11469-018-9912-x

You could basically replicate their analysis, while they measured the associations between parent-child relationship (although in simplistic manner), self-esteem and IA.

- (6) The sentences on row 358-362 require clarification/rewriting. How findings can be “novel” and at the same time “consistent with previous studies”?

- (7) When you discuss the effect of years spent at university (“seniority of students”), you should take into account the associations between IA and age, rather than hypothesize about (i) the effect of parental supervision (that you did not measure), (ii) differences in online activities between junior and senior students (which you could analyze but you did not).

- (8) The paragraph on the effect of parental supervision (which is actually parental warmth/support) needs to be completely rewritten and accompanied by relevant sources.

Limitations:

- (1) You should state response rate (row 394).

- (2) Your main predictor (type of online activities) was measured in a quite simplistic manner – yes/no. It would be more advantageous to see the proportion of these activities for each student, or the time they spent by each activity. You should clearly acknowledge this in limitations. Also, some analyses that would show (1) whether some online activities tend to combine and (2) whether the combined use bear the greater risk of IA would strengthen the manuscript. Moreover, you did not assess two of three online activities with greatest risk of IA – watching online pornography and online gambling. This should be also mentioned in limitations.

The whole manuscript needs a language editing. You should also check some recommendations (e.g. by American Psychological Association) for authors, e.g. result should be described using past tense etc.

Minor issues

Please consider following adjustments:

[38] “students log in to the internet.” => “…students log in to the internet for.” Or reformulate (e.g., …students are engaging in.”)

[39] “had internet addicted behavior” => “showed symptoms of IA”

[47] and [321] “were internet addicted users” => “showed symptoms of IA”

[48] “Online activities” => “The type of online activity”

[230] “online game” => “online gaming”

[248] “duration in the university” => “years spent at the university”

[313] “predicti59ng” => “predicting”

[408] “students log in to the internet” => “students are doing online”

I hope you find the recommendations helpful.

Signed

Katerina Lukavska

6. PLOS authors have the option to publish the peer review history of their article (what does this mean?). If published, this will include your full peer review and any attached files.

Reviewer #1: No

Reviewer #2: No

Reviewer #3: Yes: Katerina Lukavska

---

## [Author Response · Author response to Decision Letter 0]

30 Apr 2020

The comments given by both the editor and reviewers were very much constructive. We learned a lot from the comments and took all comments into consideration while revising the manuscript. Really, we appreciated their commitments and effort exerted to help us improve the manuscript.

---

## [Decision Letter · Decision Letter 1]

3 Jun 2020

PONE-D-20-00961R1

Online activities as risk factors for Problematic internet use among students in Bahir Dar University, North West Ethiopia: A hierarchical regression model

PLOS ONE

Dear Dr. Asrese,

Thank you for submitting your manuscript to PLOS ONE. After careful consideration, we feel that it has merit but does not fully meet PLOS ONE’s publication criteria as it currently stands. Therefore, we invite you to submit a revised version of the manuscript that addresses the points raised during the review process.

I consider that the manuscript has improved significantly in the current version, but it needs some important improvements, including a careful language editing. There are still some concepts, definitions and statistical issues that must be clarified. 

We look forward to receiving your revised manuscript.

Kind regards,

Alfonso Rosa Garcia

Academic Editor

PLOS ONE

Reviewers' comments:

Reviewer's Responses to Questions

**Comments to the Author**

1. If the authors have adequately addressed your comments raised in a previous round of review and you feel that this manuscript is now acceptable for publication, you may indicate that here to bypass the “Comments to the Author” section, enter your conflict of interest statement in the “Confidential to Editor” section, and submit your "Accept" recommendation.

Reviewer #1: (No Response)

Reviewer #2: All comments have been addressed

Reviewer #3: (No Response)

2. Is the manuscript technically sound, and do the data support the conclusions?

Reviewer #1: Partly

Reviewer #2: Yes

Reviewer #3: Partly

3. Has the statistical analysis been performed appropriately and rigorously? 

Reviewer #1: Yes

Reviewer #2: Yes

Reviewer #3: Yes

4. Have the authors made all data underlying the findings in their manuscript fully available?

Reviewer #1: No

Reviewer #2: Yes

Reviewer #3: Yes

5. Is the manuscript presented in an intelligible fashion and written in standard English?

Reviewer #1: Yes

Reviewer #2: Yes

Reviewer #3: No

6. Review Comments to the Author

Reviewer #1: The authors have not addressed my comments fully (The numbering of the following points are in accordance with my previous reviewer report).

1. Adhering to a conceptual framework is good, but a specific explanation of why the included variables may interplay in relation to the outcome is still needed.

2. Addressed.

3. I was suggesting that this work was not a prevalence study. Why was a sample size calculation method for prevalence study used?

4. Addressed.

5. Addressed.

6. Addressed.

7. I was not questioning the sequence of variables being entered in the model. I was saying that neither the effect size nor the p-value should be taken as indicating relative importance between variables.

8. Same as above.

Reviewer #2: The authors revised their draft well, however, the term 'internet addiction' is still used in the draft. Please check the consistency of terminology.

Reviewer #3: I would like to thank authors for their responses on my concerns raised in the first review and for the changes they made.

In my view, the manuscript has been significantly improved, especially the introduction section, where authors are much clearer about their theoretical background (Problem Behavior Theory) and thus their aims/hypotheses are now better understandable. However, there are still issues that need to be addressed before considering publications. The main problems of manuscript concern methods – sampling procedure and Internet addiction test (and its cut-off scores). Also, the manuscript requires careful language editing. Detail description of major and minor issues follows.

Methods

Sampling

• Although sampling is described in detail, some important facts are still missing. Do authors know, how many students did not meet all inclusion criteria (and thus excluded/not invited to participate)? It is usual to report the number of “eligible” participants (in this case, the number of students in selected departments) and then to report how much of them were excluded (both before and after data collection). Given to the fact, that authors report PIU prevalence, it seems important to know at least how many students were excluded because they were not using internet regularly for at least six months (one of the inclusion criteria). If the number of these students is high, then the data on prevalence of PIU among university students are most certainly biased. At any case, authors should clearly state, that the data on prevalence concern “regular internet users”, not all students.

Measures

• The assessment of peer pressure is still not clear to me. Is it that the higher score indicates higher vulnerability to peer pressure? But originally, the tool measured the resistance to peer pressure, so the scoring is different (reversed) than in original?

• Internet addiction test: I still believe that the prevalence of PIU should be based rather on cut-off score 80, than 50, while 50-79 reflects “frequent problems” and PIU should reflect rather “severe problems”. If you want to use “the milder cut-off” please find a support for this in current studies. One study (Ref. no.48) is not enough. Moreover, even this study distinguished between “severe” and “mild” addiction: “The overall prevalence of Internet Addiction was 26.50%, with severe addiction being 0.96%.”). So, at the very least, you should also report “mild” and “severe” PIU by use of mild (50) severe (80) cut-offs. I totally understand that you want to use mild cut-off for the sake of statistical analyses (it works better to compare similarly large groups…) BUT it is problem when you report prevalence… So, I really think that you should at the very least report the prevalence of PIU on the basis of strict cut-off score (80 in IAT) alongside with the prevalence based on the mild cut-off score.

• Also, IAT is rather old instrument and does not reflect well all PIU symptoms. Please, provide detailed description of how symptoms measured by IAT differ from more developed current instruments.

Results

• What are AOR? Please explain in the text.

Discussion

• Prevalence of PIU – While prevalence of PIU is one of the key findings of this study, it is important while discussing the prevalence with other studies, to include also information about measurement tools (and cut-off scores) used in reported studies. Also, it is necessary to be extra cautious when assessing prevalence based on self-reports – see Maraz, A., Király, O., & Demetrovics, Z. (2015). The diagnostic pitfalls of surveys: if you score positive on a test of addiction, you still have a good chance not to be addicted. A response to Billieux et al. 2015. Journal of Behavioral Addictions, 4(3), 151-154. And again, the prevalence of severe PIU (using strict cut-off score) should be given alongside with the prevalence of mild PIU.

• The effect of online activities (applications) on PIU. You revealed that the strongest predictor is social networking and online gaming. I wonder, whether these effects are independent on gender. According to some scholars, social networks can be overused by females, while games by males (see e.g., Koning, I. M., Peeters, M., Finkenauer, C., & Van Den Eijnden, R. J. (2018). Bidirectional effects of Internet-specific parenting practices and compulsive social media and Internet game use. Journal of behavioral addictions, 7(3), 624-632.) Could you please provide the relevant analyses (at least chi-square test of association for the frequency of gaming and social networking in male and female students)?

Limitations

• Response rate is high enough, but I am not sure, whether is accurate – see earlier points (how many students were not included because they did not meet all inclusion criteria, e.g., not being regular user of internet for at least 6 months).

• Limitations of Young’s Internet addiction test should be mentioned, please consult the current sources on PIU measurement.

Minor issues

[105] …age… please be more specific – e.g., being between XX-XX years of age or something similar

[116] …as most studies on risk factors have been conducted in western societies… Actually, most studies on PIU were conducted in Asia.

[121] …online activity variables significantly predict PIU of students… Maybe this could be better explained… “the type of online activity in which students are engaged significantly predicts his/her PIU… or something similar

[177] Online internet application is a confusing term… Maybe stick to “online activities” as you use the term in introduction

[table 1] …Secondary school… -> At secondary school (same as “at university”…)

[332] …have PIU behaviors… -> showed symptoms of PIU (and I recommend to use …showed symptoms of mild PIU)

[334] …The result in the current study, on the other hand, is higher… -> The prevalence in the current study…

[343] …Mode… -> Model 1 (?)

[368] … (decease the chance by 5.20%) -> Please report odds ratio instead.

There are still many mistypes and grammar errors. The text should be edited by native English speaker. Also, authors should carefully check whether they use common scientific terms.

[112] …are found… -> have been found

[130] …categorization of variable… -> variables

[149] …were considered to participate in the study… -> were asked to participate

[150] …representative respondents… -> representative sample

[275] …online game… -> online gaming

[355] …PIUP… -> PIU

[382] …students were more likely have PIU behavior -> were more likely to show PIU symptoms (?)

7. PLOS authors have the option to publish the peer review history of their article (what does this mean?). If published, this will include your full peer review and any attached files.

Reviewer #1: No

Reviewer #2: No

Reviewer #3: Yes: Katerina Lukavska

---

## [Author Response · Author response to Decision Letter 1]

15 Jul 2020

Responses to Reviewers 

To the Editor

Thank you for giving us this chance to revise and submit the manuscript. We have gone through all the comments of the reviewers. Many of the changes /improvements are highlighted yellow. 

Reviewer 1

Thank you very much for the comments. We have gone through the comments and improved the document accordingly.

1. The interplay between independent variables and dependent variables needs to be explained.

Response: Variables entered in the analyses were selected based on the existing literature describing possible relationship with the dependent variable. Some authors reported that the individual and environmental factors (age, sex, academic achievement, parental support, peer pressure self-esteem) significantly related to PIU. Others differentiated between dependence to the internet and dependence on the internet and argued that majority of individuals with PIU use the internet as a medium of online activities, such as social networking, online entertainments, and gaming. And these variables are suggested as more important for PIU. These are explained in the manuscript (lines 104 – 112). 

2. Concern related to sample size calculation: Why was a sample size calculation method for prevalence study used?

Response: The main purpose of the study was to learn the roles of online activities on the internet in predicting PIU. We need to know whether there was PIU and assess which variables (the control or predictor variables) better explain he variance of this phenomenon. Thus, we used the approach to estimate the sample size. 

3. Either the effect size or the p-value should be taken as indicating relative importance between variables. 

Response: As it is suggested, reporting effect size or lower p-value indicates the relative importance of the independent variables in explaining the dependent variable. In the current study, use of the internet for social networking, online gaming, peer pressure, and grade point average had lower p-vales. These variables are relatively important in explaining PIU among students. Nevertheless, the interest in this study was to assess whether socio-demographic variables (controls) or the online activities (predictors) better explain PIU among students. The interest was to compare the models in explaining the outcome variable. Our results revealed statistically significant improvement in predicting PIU of students with the online activity variables after controlling the socio-demographic background variables. 

Reviewer 2

Thank you very much for the comments. We have gone through the comments and improved the document accordingly.

1. Concern related to consistency of using terminology: using PIU instead of internet addiction. 

Response: The manuscript was edited as to the comment. 

Reviewer 3. 

Thank you very much for your critical review and thoughtful recommendations. We learned a lot from your suggestions. 

Methods section

1. Sampling: Concern related to knowing eligible participants based on inclusion/exclusion criteria and the response rate

Response: There were 2010 students in the selected 10 departments. The sample size was proportionally allocated to number of students in each department and selected students were approached through section mentors. Written consent of individual participants was obtained after being fully informed of the study purpose and procedures. The inclusion criteria (reside in the university dormitory, used internet at least for six months, and not diagnosed with anxiety or depression disorders for a year) were communicated while students were completing consents. Fortunately, no student reported discordant information to be excluded from the study. Thus, all were eligible to participate in the study. During data cleaning, we identified that 32 students did not fully complete questionnaire, hence excluded from the study. Data obtained from 812 respondents (96.2% response rate) were reported in this study. 

2. Measurement 

A. Clarity on the assessment of peer pressure

Response: Originally, the tool was developed to measure resistance to peer pressure. In this study, the scoring is reversed: higher score indicated higher vulnerability to peer pressure. 

B. Cut-offs used for Young’s Internet addiction test scores?

Response: The cut-offs for Internet addiction test scores were modified as to the reviewer’s comments. 

Discussion section

A.Concern related to prevalence of PIU.

Response: In the revised manuscript, the prevalence of PIU was discussed in line previous research reports those used similar measurement tool (Internet Addiction Test) as suggested by the reviewer (lines 331 – 339). 

B. Concern related to whether the effects of online activities on PIU was independent of gender (chi-square test of association for the frequency of gaming and social networking in male and female students)

Response: In the revised manuscript, bivariate analysis of our findings revealed that female students were more likely to show symptom of PIU than their male counterparts. When the effects of other variables are controlled in the multivariate analysis, such difference failed to reach significant. The main purpose of the study was to learn the roles of online activities on the internet in predicting PIU. Gender was one of the control variables entered first in the regression model and the improvement of the second model with online activity variables in explaining the variance in PIU was assessed. Below is the distribution of male and female students by online activities assessed in the study.

Online activities Sex 

χ2

 Male

n(%) Female

n(%) 

Social networking use 406(75.9) 207(74.7) .132

No use 129(24.1) 70(25.3) 

Entertainment use 408(76.3) 190(68.6) 5.531*

No use 127 (23.7) 87(31.4) 

Academic works use 393(73.5) 183(66.1) 4.838*

No use 142(26.5) 94(33.9) 

Online game use 119(22.2) 56(20.2) .443

No use 416(77.8) 221(79.8) 

Limitation section

1. Concern related to limitations of Young’s Internet addiction test should be mentioned.

Response: The revised manuscript is improved as to the comments given by the reviewer (lines 416 – 423).

 Minor issues

Language and editorial works

Response: A professor in English language teaching English as a foreign language edited the manuscript.

---

## [Decision Letter · Decision Letter 2]

12 Aug 2020

PONE-D-20-00961R2

Online activities as risk factors for Problematic internet use among students in Bahir Dar University, North West Ethiopia: A hierarchical regression model

PLOS ONE

Dear Dr. Asrese,

Thank you for submitting your manuscript to PLOS ONE. After careful consideration, we feel that it has merit but does not fully meet PLOS ONE’s publication criteria as it currently stands. Therefore, we invite you to submit a revised version of the manuscript that addresses the points raised during the review process.

We ask you to review some minor points in the manuscript, that I will revise alone (I will not send the new version to reviewers). In particular, following the comments of reviewer 1, you should consider to eliminate the word "relative" in the first line in the "Conclusion". You also should try to fix the consistency mistakes identified by reviewer 3.

We look forward to receiving your revised manuscript.

Kind regards,

Alfonso Rosa Garcia

Academic Editor

PLOS ONE

Reviewers' comments:

Reviewer's Responses to Questions

**Comments to the Author**

1. If the authors have adequately addressed your comments raised in a previous round of review and you feel that this manuscript is now acceptable for publication, you may indicate that here to bypass the “Comments to the Author” section, enter your conflict of interest statement in the “Confidential to Editor” section, and submit your "Accept" recommendation.

Reviewer #1: (No Response)

Reviewer #2: All comments have been addressed

Reviewer #3: All comments have been addressed

2. Is the manuscript technically sound, and do the data support the conclusions?

Reviewer #1: Partly

Reviewer #2: Yes

Reviewer #3: (No Response)

3. Has the statistical analysis been performed appropriately and rigorously? 

Reviewer #1: Yes

Reviewer #2: Yes

Reviewer #3: (No Response)

4. Have the authors made all data underlying the findings in their manuscript fully available?

Reviewer #1: No

Reviewer #2: No

Reviewer #3: (No Response)

5. Is the manuscript presented in an intelligible fashion and written in standard English?

Reviewer #1: Yes

Reviewer #2: Yes

Reviewer #3: (No Response)

6. Review Comments to the Author

Reviewer #1: 1. Listing the previously identified risk factors does not mean explaining the interplay between them.

2. The authors have conducted a logistic regression. Sample size calculation should be based on this.

3. I repeat: the way the authors do the analysis, the relative importance cannot be quantified. So claiming that they have assessed relative importance between the variables is misleading.

Reviewer #2: I recommend for the publication in the current form.

I hope that more interesting studies will come out from your region in regard to addictive behaviors related to new technologies.

Reviewer #3: Thanks for the opportunity to review this manuscript. I believe it is a solid piece of research in its current form. Only few minor and rather "technical" issues should be corrected (e.g. past sense while describing results - rows 251, 252 and elsewhere, consistency in replacement of IA by PIU - row 450).

7. PLOS authors have the option to publish the peer review history of their article (what does this mean?). If published, this will include your full peer review and any attached files.

Reviewer #1: No

Reviewer #2: No

Reviewer #3: **Yes: **Katerina Lukavska

---

## [Author Response · Author response to Decision Letter 2]

12 Aug 2020

Responses to Reviewers 

To the Editor

Thank you for giving us this chance to revise and submit the manuscript. We have gone through the comments of the reviewers. The changes /improvements are highlighted yellow. 

Reviewer 1

Thank you very much for the comments. We have gone through the comments and improved the document accordingly.

1. You should consider eliminating the word "relative" in the first line in the "Conclusion 

Response: we changed as to the comment.

Reviewer 3. 

Thank you very much for your critical review and thoughtful recommendations. We learned a lot from your suggestions. 

1. Technical issues should be corrected (e.g. past tense while describing results - rows 251, 252 and elsewhere, consistency in replacement of IA by PIU - row 450).

Response: Improvements are made as to the comments.

---

## [Editor Report · Decision Letter 3]

25 Aug 2020

Online activities as risk factors for Problematic internet use among students in Bahir Dar University, North West Ethiopia: A hierarchical regression model

PONE-D-20-00961R3

Dear Dr. Asrese,

We’re pleased to inform you that your manuscript has been judged scientifically suitable for publication and will be formally accepted for publication once it meets all outstanding technical requirements.

Kind regards,

Alfonso Rosa Garcia

Academic Editor

PLOS ONE
---

## [Editor Report · Acceptance letter]

31 Aug 2020

PONE-D-20-00961R3 

Online activities as risk factors for Problematic internet use among students in Bahir Dar University, North West Ethiopia: A hierarchical regression model 

Dear Dr. Asrese:

I'm pleased to inform you that your manuscript has been deemed suitable for publication in PLOS ONE. Congratulations! Your manuscript is now with our production department. 

Kind regards, 

on behalf of

Dr. Alfonso Rosa Garcia 

Academic Editor

PLOS ONE